# Clonal Architecture and Evolutionary Dynamics in Acute Myeloid Leukemias

**DOI:** 10.3390/cancers13194887

**Published:** 2021-09-29

**Authors:** Matthieu Duchmann, Lucie Laplane, Raphael Itzykson

**Affiliations:** 1Génomes, Biologie Cellulaire et Thérapeutique U944, INSERM, CNRS, Université de Paris, 75010 Paris, France; matthieu.duchmann@inserm.fr; 2Laboratoire d’Hématologie, Hôpital Saint-Louis, Assistance Publique-Hôpitaux de Paris, 75010 Paris, France; 3Institut d’Histoire et Philosophie des Sciences et des Techniques UMR 8590, CNRS, Université Paris 1 Panthéon-Sorbonne, 75010 Paris, France; lucie.laplane@gustaveroussy.fr; 4Gustave Roussy Cancer Center, UMR1287, 94805 Villejuif, France; 5Département Hématologie et Immunologie, Hôpital Saint-Louis, Assistance Publique-Hôpitaux de Paris, 75010 Paris, France

**Keywords:** acute myeloid leukemia, clonal heterogeneity, evolutionary dynamics, drug resistance, prognosis

## Abstract

**Simple Summary:**

Acute myeloid leukemias (AML) results from the accumulation of genetic and epigenetic alterations, often in the context of an aging hematopoietic environment. The development of high-throughput sequencing—and more recently, of single-cell technologies—has shed light on the intratumoral diversity of leukemic cells. Taking AML as a model disease, we review the multiple sources of heterogeneity of leukemic cells and discuss the definition of a leukemic clone. After introducing the two dimensions contributing to clonal diversity, namely, richness (number of leukemic clones) and evenness (distribution of clone sizes), we discuss the mechanisms at the origin of clonal emergence and the causes of clonal dynamics including neutral drift, and cell-intrinsic and -extrinsic influences on clonal fitness. After reviewing the prognostic role of leukemic diversity on patients’ outcome, we discuss how a better understanding of AML as an evolutionary process could lead to the design of novel therapeutic strategies for this disease.

**Abstract:**

Acute myeloid leukemias (AML) results from the accumulation of genetic and epigenetic alterations, often in the context of an aging hematopoietic environment. The development of high-throughput sequencing—and more recently, of single-cell technologies—has shed light on the intratumoral diversity of leukemic cells. Taking AML as a model disease, we review the multiple sources of genetic, epigenetic, and functional heterogeneity of leukemic cells and discuss the definition of a leukemic clone extending its definition beyond genetics. After introducing the two dimensions contributing to clonal diversity, namely, richness (number of leukemic clones) and evenness (distribution of clone sizes), we discuss the mechanisms at the origin of clonal emergence (mutation rate, number of generations, and effective size of the leukemic population) and the causes of clonal dynamics. We discuss the possible role of neutral drift, but also of cell-intrinsic and -extrinsic influences on clonal fitness. After reviewing available data on the prognostic role of genetic and epigenetic diversity of leukemic cells on patients’ outcome, we discuss how a better understanding of AML as an evolutionary process could lead to the design of novel therapeutic strategies in this disease.

## 1. Introduction

For several decades, leukemias as other cancers have been envisaged as evolutionary processes [1,2]. From cancer initiation to diagnosis and progression, leukemic cells acquire genetic and epigenetic alterations. As single-cell organisms reproducing asexually through cell division, leukemic cells follow evolutionary processes such as selection and drift, called clonal evolution in cancer. These processes shape the architecture of leukemia by modulating the survival and expansion of each leukemic cell population. Evolution of cancer cells plays a major role in cancer development and treatment response; thus, it is a topic of central importance. Focusing on acute myeloid leukemias (AML), we sequentially address three questions: Which traits should form the basis of clone identity? What information does assessment of clonal architecture provide on the underlying evolutionary processes? How can this information instruct the choice of therapy for AML patients?

## 2. Clonal Identity in AML

In the cancer field, the term ’clone’ has been used to group cells sharing traits inherited from a common ancestral cell. Cells within a clone are considered to have the same fitness, i.e., the same competitive advantage in a given environment, compared to other clones and to normal cells. However, cells are characterized by multiple biological traits, some of which are highly context-dependent [3], raising the question of the choice of biological properties used to define clonal identity. Ideally, clonal identify should be based on traits that best explain the evolutionary trajectories of cancer cells.

### 2.1. Genetic Identity

Owing to a long tradition of genetic reductionism in the field of cancer biology, and technological limitations to study other traits, clonal identity has long been based on genetic alterations [4]. With aging, hematopoietic stem and progenitor (HSPC) cells accumulate random genetic alterations [5,6,7]. Even after transformation, leukemic cells still acquire new cytogenetic alterations and point mutations, and virtually each cell harbors unique, ‘private’, genetic alterations [1,5]. Most of these mutations are in noncoding regions or have little impact on protein function and are deemed neutral for cell function [8]. Conversely, a minority of these alterations can provide a fitness advantage to the cells, leading to clonal expansion [8]. These high-impact driver mutations include alterations that increase self-renewal, proliferation, or survival, and in vitro and in vivo experiments have shown that they are important for leukemic initiation and progression. As these drivers are believed to strongly influence the transformation of hematopoietic into pre- and fully leukemic cells, they are attractive traits to determine clonal identity. However, classifying a mutation as a driver or a passenger is not always straightforward, and relies on current scientific knowledge, in silico predictions, and known epistatic interactions. Recent studies suggest that the fitness impact of a mutation is positioned on a continuum [9]. Some variants might provide only a limited fitness advantage (‘mini-drivers’) [10], while others might be deleterious. Yet, all these variants could contribute to shape clonal evolution in leukemias. 

Beyond the mutational landscape at the bulk level, the precise combination of mutations occurring in each single cell is important to define clonal identity. This information has been available for a long time for cytogenetic alterations [1,2,11,12,13], although conventional karyotyping has low resolution and may be subject to culture bias. For point mutations, clonal composition has been inferred from whole-genome sequencing (WGS) of bulk samples by clustering variants with similar allelic frequencies (VAF) [14,15], a procedure that does not resolve complex architectures [16]. Exome sequencing or targeted sequencing of panels including at least 50–100 genes recurrently mutated in AML can capture partial information on clonal structure. These approaches often compensate their limited coverage compared with WGS with greater depth, allowing more precise assessment of VAFs and more robust clustering of the variants captured [5,17,18,19,20]. Single-colony [5,6,21], plate-based single cell [22,23], and more recently, high-throughput, droplet-based genotyping protocols [24,25] have contributed to describe the exact clonal composition of AML (Figure 1). These studies revealed the co-occurrence of variants in genes requiring biallelic inactivation (i.e., *TET2*, *DNMT3A*), or on the other hand, branched architectures where multiple mutations in the same oncogenic pathways occurred in different cells. Oncogenes with at least a partially overlapping function in AML include receptor tyrosine kinase and RAS-MAPK genes (*FLT3*, *KIT*, *NRAS*, *KRAS*, etc., hereafter signaling genes) [26], *TET2* and *IDH1/2* [27], or *TP53* and *PPM1D* [28]. Finally, these studies identified some co-occurring mutations linked to clonal dominance, suggesting a strong cooperative interaction between them, such as for *DNMT3A* and *IDH1/2* mutations [24,25].

### 2.2. Functional Identity

In AML, only few recurrent cytogenetic and gene alterations can predict responses to chemotherapy at the population level [29,30], and robust individual prediction of chemosensitivity remains unsatisfactory [31,32]. AML cells are composed of different subpopulations expressing different transcriptional programs [33], intracellular signals, and surface protein expressions [34]. Some of these differences reflect incomplete blockage of differentiation, with some leukemic cells resembling normal hematopoietic progenitors and mature cells (Figure 1) [33,34]. These intratumoral differences are clinically relevant, as a high-burden of immature blasts with the long-term self-renewal property has been associated with chemoresistance and poorer prognosis in multiple patient cohorts [34,35,36], whereas mature monocyte-like cells have immunosuppressive capacity [33] and are associated with resistance to the recently approved BCL-2 inhibitor venetoclax [37,38]. 

Functional heterogeneity is sustained by an epigenetic heterogeneity [39,40]. Indeed, epigenetic alterations have been reported in AML, only some of which are associated (and potentially driven) by specific genetic alterations [40,41]. Similar to genetic alterations, there are various degrees of intratumor epigenetic heterogeneity in AML [39,40,42]. Increased methylation diversity at diagnosis is distinct from genetic heterogeneity and is associated with higher risk AML and poorer survival [39,40]. Epigenetic heterogeneity is also dynamic during AML evolution, and subject to evolutionary processes such as selection or drift, which may be dissociated from the evolution of genetic heterogeneity [39].

Functional heterogeneity might be also explained by non-(epi)genetic, stochastic, multigenerational fluctuations. Such pervasive ‘biological noise’ [43,44], whose underlying mechanisms remain elusive, has been associated with in vitro chemoresistance in cancer [44,45] and AML [46] cells. In solid cancers, resistance to targeted therapies can emerge from multiple independent evolutionary paths explored by cells after first entering a persistent state [47,48]. Such a phenomenon, where plasticity first allows the individuals (here, cancer cells) to explore the phenotypic space before adaptive traits (e.g., drug resistance) are enforced by heritable genetic or epigenetic alterations, is reminiscent of the ‘Baldwin effect’ in evolutionary biology [49]. A similar phenomenon has recently been described in childhood B-cell precursor acute lymphoblastic leukemia [50]. Indeed, transient changes involved in drug resistance have been reported to occur in vivo [51] and ex vivo [52], which may allow a subset of AML cells to survive long enough time to acquire heritable resistance (epi)mutation [43].

### 2.3. Multi-Modal Identity

Genetic and functional identities are interconnected. At the population level, AML oncogenic subgroups are associated with specific transcriptomic programs [53]. Subclonal mutations contribute to intratumor functional heterogeneity. Patient-derived xenograft (PDX) experiments have reported that genetic clones have distinct engraftment potentials [22,24,54,55], although engraftment capacity was highly dependent on the experimental context, and poorly predicted clonal architecture at relapse [22,55]. Genotyping of sorted subpopulations [22] and, more recently, single-cell multiomics technologies [24,25,33,56] highlighted that some variants are enriched in clusters of leukemic cells defined by specific RNA [33,56] or protein [24,25] expression patterns. Whether these mutations directly activate a differentiation program or confer a fitness advantage only at a specific differentiation stage has yet to be investigated. A pan-cancer analysis studying gene-expression heterogeneity through multitask evolution theory has shown that driver mutations tune gene expression towards specialization in certain tasks [57]. Emerging single-cell methodologies integrating genotyping, gene expression, and protein expression such as TARGET-seq are promising for studying these intricate relationships, and to define multimodal clonal identities (Figure 1) [58]. 

## 3. Origin of Intraleukemic Heterogeneity and Clonal Evolution

Each AML results from a unique composition of clones cohabiting and/or succeeding each other throughout the natural history of the disease. Behind the uniqueness of each patient’s AML, various modes of evolution can be distinguished. 

### 3.1. Characterizing Intratumor Heterogeneity

Building on ecology theories, we can first stratify diversity in AML according to spatial scale. Global intraleukemic heterogeneity can be compared with the gamma diversity of an ecosystem, which represents the diversity at the scale of several habitats (Figure 2) [59]. Gamma diversity has two components: heterogeneity within a specific sampling site (alpha diversity, Figure 2) and differences across all leukemic sites (beta diversity, Figure 2). The difference between alpha and beta diversities is less clear in AML than in solid cancers, but some discrepancies in the genetic makeup of bone marrow leukemic cells and myeloid sarcomas have been reported, suggesting incomplete mixing of leukemic cells and, thus, the existence of beta diversity [60]. Furthermore, intravital imaging studies have shown that chemoresistant AML cells have a decrease in motility [61], a phenomenon that could generate local differences in clonal composition between leukemic niches at relapse. The alpha diversity itself has two components: the number of clones (clonal richness) and their relative abundance (clonal evenness, Figure 2). Different indices exist to quantify alpha diversity [62]. The most frequently used are the number of clones [12,25], which is sometimes approximated by counting the number of mutations in the most recurrent AML oncogenes [17,63], Shannon’s index [17,24,25], and Simpson’s index [17]. 

### 3.2. Source of Clonal Richness 

Given that clonal richness is a fuel for clonal evolution, it is crucial to understand its determinants. If we apply to cancer the conceptual framework of evolutionary dynamics in asexual organisms, most of which has been fueled by experiments on bacteria, we can predict that the mutational rate, the number of generations from initiation to diagnosis, and the effective population size are the key determinants of clonal richness in leukemia (Figure 3) [64,65]. Genetic instability is believed to be limited in AML, as point mutation and cytogenetic alteration burden is in a lower range compared with other cancers, although the variance between AML patients is particularly high [8,66,67]. Rare patients with germline mutations in DNA repair genes such as patients affected by Xeroderma Pigmentosum C syndrome showed an increase in the number of mutation rates in AML [68]. In most AML cases, the mutational signatures found are characterized by C > T transitions, which are linked to spontaneous deamination of 5-methyl cytosine and are related to aging [5,67,69]. Thus, it is likely that most mutations have accumulated before initiation of the leukemia, as most of them are present in all AML cells in the sample, and there is a high correlation between the total number of mutations and the patient’s age [5]. Even when focusing on recurrent driver mutations, higher clonal diversity correlates with older patient age [17,24]. Of note, patients with clinically or molecularly defined secondary AML (sAML), who are usually older than de novo AML patients, have an increased number of genetic drivers [70,71]. Thus, the number of generations is important, especially in the case of neutral evolution, which is frequent in solid cancers [72] but may also exist in AML [73]. Mutation rate can be dynamic over time, and while most mutations accumulate slowly and gradually, there are cases of punctuated evolution with the acquisition of large genetic rearrangements, such as chromothripsis [74], especially in patients with genetic instability such as *TP53* inactivation. Mutation rate can also transiently increase as an adaptive phenomenon in the context of therapeutic selection pressure [75]. Finally, the effective population size, defined as the number of individuals actively contributing to evolution in the census population, is an important determinant of clonal richness because the number of new mutations in each generation scales with this parameter [64,76]. Determining effective population size in AML is not trivial. Hierarchical renewing tissue organization is a protection against somatic mutation fixation by limiting the number of cells with self-renewing capacity, reducing their number of divisions, and purging mutations through differentiation [77,78]. In AML, the effective units of selection are the minority of cells able to proliferate with a long-term self-renewal capacity, namely, leukemic stem cells (LSCs) [79]. PDX experiments have shown that most AML clones at diagnosis can engraft in immunocompromised mice, and thus, are sustained by a population of self-renewing cells [22,24]. Studies on solid cancers have shown that the expression of stemness signatures is associated with increased genetic diversity [80]. Similar studies in AML are awaited. Whether stemness is a fixed trait or a more adaptative state subject to equilibrium is still a matter of debate and investigation [81], including in AML [82]. In solid tumors, accumulating evidence suggests that dedifferentiation of cancer nonstem cells can contribute to the regeneration of the cancer stem cell pool [83,84]. A recent study has reported reversibility of differentiation in a murine model AML driven by inducible suppression of endogenous PU.1 [85]. The potential contribution of such a phenomenon to the stemness capacity and long-term evolution of AML has yet to be investigated. 

### 3.3. Clonal Evolution Processes

With equal richness, clonal evolution can follow different paths, leading to differences in relative clonal abundance. A wide number of biological phenomena can account for these differences in clonal evenness. First, although clonal evolution is often conceived as a Darwinian process of evolution by natural selection, it is important to note that neutral evolution, which is widespread in the evolution of organisms, can also occur in cancer. Indeed, recent data suggest that neutral evolution is frequent after initial transformation in some solid cancers [72,73] and may also exist in AML [73,86]. In this model, cancer cells accumulate alterations during the early steps of transformation, whereas clonal sweeps leading to expansion and fixation of adaptive clones at late stages of progression are rare. Depending on the tissue structure and organization, neutral evolution can lead to high clonal richness with relatively good evenness (clone size is proportional to clone age), or to clonal dominance due to genetic drift. In spatially structured populations, small niches favor clonal sweeps and tumor homogeneity, while segregated regions limit clonal sweep, thus, favoring diversity between regions (low alpha diversity in regions but high beta diversity when comparing regions) [87]. In leukemias, where cell mixing is supposed to be high, a small effective size (i.e., few LSCs) can also lead to random clonal expansion or extinction due to genetic drift [64,65,76,77,88]. This may explain why some mutations increasing self-renewal such as *TET2* [89] and *DNMT3A* [90] are early events in AML, increasing the probability of a secondary driver mutation that can reach fixation without extinction of the preleukemic clone due to drift. Neutral evolution can also lead to clonal expansion due to the founder effect, when some leukemic cells invade another environment, e.g., in the context of myeloid sarcomas [60].

Clonal evolution can also be a deterministic process underlined by natural selection. In this case, clonal evenness will largely depend on the fitness of each clone. If a clone acquires a fitness advantage, it will expand and outcompete other clones. Interpretation of fitness is always relative and should account for the fitness of normal hematopoietic cells, which may decline with age, potentially explaining the age-dependence of some subsets of acute leukemias [91]. Regarding AML, the age-related decline in HSC function—despite their increased number [92,93,94]—might favor the expansion of preleukemic HSCs with better self-renewal capacities [89,90], resulting in age-related clonal hematopoiesis (ARCH) and, eventually, AML transformation. In sAML patients, MDS or MPN founding clones are still detectable at the AML phase, although sometimes outcompeted by a minor subclone leading to a clonal sweep [70,95]. Some specific drivers, notably in signal transduction genes, have been significantly associated with MDS progression to sAML [95].

Furthermore, fitness is often envisaged as a cell-intrinsic trait of cancer cells. However, fitness of a clone may depend on its relative abundance because its expansion depends on a signal, notably, a diffusible factor (e.g., a growth factor, metabolite, or exosome) available in a limited amount (density-dependent fitness) secreted by the clone itself or by another clone [96]. The potential paracrine effect of the oncometabolite R-2-hydroxyglutarate secreted by *IDH* mutant clones may exemplify such ‘public good’ games in AML [97]. Fitness of a cancer cell may also depend on a pairwise dialogue with a neighboring cell. Such a process, called cell competition, has been extensively studied in the context of normal multicellular development and relies on intercellular dialogues between adjacent cells to eliminate cells with lower fitness, while preserving the tissue architecture [98]. In solid tumors and T-cell acute leukemias, cell competition has been shown to prevent transformation by eliminating early-stage clones [99,100]. Conversely, cancer cells may take advantage of this process, particularly with tissue aging. Recent studies suggest that breast and colon cancers may hijack these developmental processes to eliminate surrounding stromal cells [101]. A p53-dependent mechanism akin to cell competition has been shown to contribute to the resilience of the murine hematopoietic system upon radiation-induced DNA damage [102]. Leukemic and normal HSPC have been shown to compete for the bone marrow niche [103,104], which are saturable ecosystems important for both HSC [105,106,107,108] and LSC functions [103,104]. Leukemic cells may even remodel the bone marrow environment to preferentially support leukemic growth at the expense of normal HSPCs [109,110,111,112,113]. Cytokines are important resources for leukemic cells, and some genetic alterations might provide cytokine hypersensitivity or independence [114]. PDX experiments using immunodeficient mice expressing humanized cytokines (NSG-SGM3) or not (NSG) reported different clonal architectures and phenotypic differentiation in the two strains [22] suggesting that clones have specific cytokine requirements for engraftment and/or expansion. In addition to this positive selection, leukemic cells are facing hazards that may jeopardize their survival. Some leukemic clones might have enhanced resistance to inflammation [115,116,117] or to therapy-induced damage [118,119]; the latter might explain the higher incidence of *TP53* and *PPM1D* mutations in therapy-related AML [119]. Sequencing of paired diagnosis/relapse samples showed that chemotherapy induces variable changes in the genetic clonal composition, with the loss of some drivers and/or expansion of others [15,19,120]. The dominant clone at relapse can derive directly from the dominant clone at diagnosis or from the expansion of a minor subclone frequently detectable at diagnosis [15,82]. In a minority of cases such as AML relapses with *NPM1* loss, genetic distance between diagnosis and relapse is more important, with only preleukemic drivers persisting at relapse. This pattern of evolution, often associated with a longer time to relapse, should lead research to consider the post-treatment leukemia as second AML rather than a true relapse [20]. Genetic mutations can convey resistance to treatment, mainly in the context of targeted therapy [18,121,122], but this mechanism might also be involved in chemoresistance [19]. Resistance to treatment can also be conveyed by a pre-existing stemness program [82,123,124] and/or activation of a senescence program [52]. Clonal evenness is, thus, likely to change when the environment changes. Notice that, in addition to deterministic fitness, stochastic biological noise can also play a role in clonal expansion under changing selection pressures, particularly in the case of changes in the fitness landscape, as discussed above [44,45].

Even when considering a fixed environment and neglecting density-dependent phenomena, inferring the fitness of clones harboring several drivers based on data for each individual driver (e.g., from syngeneic mouse models [89,90]) is not straightforward. Based on bulk sequencing data, each additional driver has been proposed to lead to a 15-20% gain in fitness [3,125]. However, gene–gene interactions (‘epistasis’) are likely to affect the combined output of combinations of driver mutations. Some combinations of mutations may synergize at the molecular level, leading to large gains of fitness and clonal dominance, strong association at the population level, and poorer prognosis, as exemplified in AML by the combination of *IDH2* and *DNMT3A* mutations [25,63,126,127]. During AML progression, fitness gains of mutations tend to be lower, owing to partially overlapping effects (diminishing-returns epistasis [64]), progressive alteration of the cellular signaling networks, and increased immunogenicity [9,128]. This phenomenon is pervasive across cancer subtypes [129], and could contribute to the persistence of clonal heterogeneity, notably, explaining why subclones with the highest number of (supposedly driver) mutations are not necessarily dominant [25,70]. The persistence of ancestor clones prior to therapy may then fuel the reservoir of resistance clones [15,25,70].

Clones are often conceived primarily as competitors, but they can also cooperate [130,131,132,133,134]. A small population can promote growth of other clones by noncell autonomous mechanisms, leading to a strong interdependency and clonal equilibrium.

Studying leukemia through the prism of multitask evolution theory helps to understand inter- and intraleukemic heterogeneity. Leukemic cells must perform several tasks to survive, but theoretically, no cell can be optimal for all tasks because of limited biomass or interference between them [3,57,135]. Therefore, leukemic cells must find trade-offs between these tasks. Spatiotemporal heterogeneity of selection pressures might increase the diversity of the cancer population, a feature more obvious in solid oncology as the tumor grows [3]. However, this might also be the case in AML, as spatial gradients of hypoxia have been reported in the bone marrow [136], and progressive saturation of bone marrow niches at advance stages of AML might lead to temporal heterogeneity, promoting the migration to peripheral blood and spleen [137]. 

## 4. Clinical Impact of Clonal Architecture

Evolution of cancer cells represents a challenge for cancer treatment. A better understanding of the underlying processes that drive clonal evolution can provide new therapeutic opportunities.

### 4.1. Impact of Clonal Architecture at Diagnosis

Clonal diversity has been associated with poorer outcomes in AML. Clonal richness, defined on the number of cytogenetic clones, has been associated with poorer prognosis, especially in AML patients with a complex karyotype [12,13]. The impact of genetic heterogeneity for point mutations is more difficult to assess. Recent technological advances allow the precise characterization of the clonal architecture at the single-cell level of large cohorts of AML patients [24,25], and the prognostic impact of clonal heterogeneity and specific clonal architectures in a uniformly treated patients cohort should be investigated soon. A higher number of driver mutations has been consistently associated with poorer prognosis in AML [17,63,138,139]. Epigenetic heterogeneity may also portend a poorer prognosis [39,40]. 

Clone size distribution and specific clonal architectures have been associated with prognosis in AML. In two independent cohorts, the presence of clonal dominance inferred from bulk sequencing was shown to predict shorter survival independently of the number of clones [17]. Interestingly, FLT3–ITD allelic ratio, which integrates both zygosity and clonal dominance, is a strong stratifying driver in AML patients. Whether homozygous state, dominance of the FLT3-ITD clone, or both, are prognostic remains to be elucidated [140]. In core-binding factor AML, branching caused by parallel evolution of clones harboring distinct signaling pathway mutations (a process reminiscent of clonal interference reported in bacteria populations under antibiotic selection pressure) is an independent predictor of outcome [26]. Clonal interference of *FLT3* internal tandem duplications also portends greater risk of treatment failure [141]. Importantly, all of these analyses have been carried out in patients treated with conventional ‘7 + 3’ chemotherapy. Whether these findings are specific to this intensive cytotoxic regimen or reflect the natural history of the disease remains unknown.

The link between clonal architecture and prognosis remains unclear in AML and has been mainly reported in studies where clonal architectures were inferred from bulk samples. Future studies defining clonal architecture based on multiple traits at the single-cell level might refine this association with outcome. Clonal architecture could play a direct, causal role on AML prognosis. Alternatively, clonal architecture could be a proxy of an underlying process itself, causing chemoresistance or relapse. Determinist mechanisms linking intraleukemic heterogeneity and chemoresistance remain elusive. Sequential driver acquisition could lead to additive fitness, ultimately resulting in an aggressive chemoresistant clone. However, lack of correlation between the accumulation of drivers in individual clones and clonal dominance in single-cell genotyping studies challenges this hypothesis [25]. Furthermore, the dominant clone at diagnosis is often not the one found at relapse [17]. Intraleukemic heterogeneity can also be considered as a reservoir of resistance. Indeed, even undetectable mutations generated with neutral evolution can later be involved in resistance and relapse, a phenomenon described as the ‘Dykhuizen-Hartl effect’ in the neutral theory of evolution [142]. Indeed, in most patients, mutations associated with resistance to targeted therapies can be detected at very low allelic frequencies in pretreatment samples [24,121]. Intratumoral heterogeneity can also be a mechanism of immune escape, as reported in solid cancers [143]. Finally, clonal interference of signaling mutations could indicate strong selection pressures on this oncogenic pathway with the successful emergence of well-adapted clones [144], or reflect clonal cooperation involved in chemoresistance [132].

On the other hand, clonal architecture might simply reflect some underlying mechanisms responsible for both clonal heterogeneity and chemoresistance. Mutation rate, number of generations, and effective population size can impact clonal architecture (Figure 3) and, thus, might provide underlying explanations of the relationship between clonal architecture and AML prognosis. Specifically, a causal role for the number of generations and, hence, longer evolutionary history, may explain the association of clonal heterogeneity with older age and sAML [12,13,17,25]. Increased stemness has been associated with a poor prognosis in most AML patients [34,35,36] and is a factor of clonal richness [80]. 

### 4.2. New Treatment Strategies

In the past decade, the development of new drugs targeting genetic [145,146,147,148] and functional [149] AML dependencies has led to significant clinical progress. To date, there is no strong evidence that the clonal or subclonal nature of these targetable hits influences response to FLT3 or IDH inhibitors in AML [145,150]. In a subset of patients, multiple targetable clones may be present. In that instance, although it may seem intuitive to target the dominant or more ancestral mutation, ‘commensal’ effects of minor subclones on nonmutated leukemic cells may represent a vulnerability for the leukemic bulk, as exemplified by *IDH* mutant clones [97,150]. Conversely, targeting an ancestral mutation may allow the expansion of rare subclones that, in absence of therapy, are encumbered by an untapped vulnerability to oncogene overdose [151]. Single-agent targeted therapy is not curative in AML. Primary resistance and relapses frequently emerge from a minor, pre-existing resistant clone expanding with therapy, though they are often barely detectable prior to therapy with current methods, as discussed above [121,152]. Interestingly, clonal selection might sometimes lead to a drug addiction phenomenon, where emerging clones are only fit under drug exposure. For instance, *NRAS/FLT3-ITD* double-mutated clones have been reported under treatment with the FLT3 inhibitor gilteritinib, whereas these mutations are usually mutually exclusive in the absence of this drug. Drug withdrawal (“drug holiday”) often induces regression of these addicted clones [121]. This illustrates the overall fitness cost of resistance, a well-known phenomenon in bacteria and cancer cells [153,154,155]. Combining or alternating drugs are two strategies borrowed from antimicrobial therapy that can effectively prevent resistance by either having deeper on-target effects, targeting different vulnerabilities, preventing signaling pathway reactivation, taking advantage of synthetic or collateral lethality, or killing different cell populations [156]. One of the most elegant strategies is to use fitness trade-offs of resistance to build an evolutionary trap, where resistance to drug A sensitizes to drug B and vice versa. This ‘double bind’ strategy exploiting antagonistic pleiotropy is starting to be systematically investigated in solid tumors and in AML [157,158].

Predicting drug sensitivity only through molecular profiling has been disappointing so far in oncology [159]. It is therefore important to predict the best available drug combinations for each patient. Ex vivo drug screening has been reported to predict drug sensitivity in AML patients [160,161]. However, the limited number of cells and short-term end point restrict testing many combinations and studying complex and heterogeneous responses on a polyclonal leukemic population. Indeed, response of the predominant subpopulation has been predicted to be insufficient to determine the best drug combination [162]. Flow-cytometry-based drug screening has shown that distinct immunophenotypic populations have different sensitivity to therapies [38], e.g., a monocytic differentiation is associated with venetoclax resistance [37,38]. In addition, readouts other than viability might be important, including differentiation and stemness [163]. Authors recently reported on the use of sc-RNAseq with single-agent ex vivo drug screening to predict the best combination at the bulk level [164], but this strategy might also be able to predict which combination therapy can be effective at the clonal level [162]. 

Other emerging strategies take advantage of selective pressures that have shaped the clonal architecture of AML at diagnosis. Clonal interference might point out a strong selective pressure on the involved oncogenic pathway, thus, representing a targetable Achille’s heel [144]. One crucial challenge is to prevent the selection and onset of a highly resistant disease. Contrary to chronic hematologic malignancies, the problem of when to start treatment to avoid selecting a chemoresistant clone versus delaying progression is not relevant in AML. Adaptive strategies assume that resistant cells are a minority at diagnosis because of the fitness cost of resistance [165]. An intensive treatment would preferentially kill sensitive cells, leading to a competition release and a relapse mostly composed of resistant cells. Adaptive strategies suggest using a less-intensive treatment sufficient to stabilize tumor size, while maintaining an effective competition between sensitive and resistant cells. Though this strategy has shown some clinical relevance in solid tumors [165,166,167,168], the requirement of complete remission to obtain a survival benefit with nonintensive therapies in AML suggest that adaptive therapy may not be relevant in this disease [169]. Other strategies less likely to lead to resistance would imply targeting interclonal cooperation between clones [62,170], or LSCs and their microenvironment [171].

## 5. Conclusions

Intraleukemic diversity can occur at various scales, from local to global, and take various shapes that can usefully be described through richness (the number of clones) and evenness (the relative size of the clones). Multiple factors contribute to clonal emergence, including mutation rate, evolutionary time, and effective population size. Clonal expansion can occur due to a fitness advantage or by random drift in a small effective population. Clonal coexistence can be caused by competition or, instead, cooperation. It may simply reflect a task repartition with even fitness advantage. Single-cell multiomics will be necessary to determine a multimodal clonal identity. A clonal architecture reconstruction solely based on genetics may provide a biased view of the true diversity of leukemic cells. For instance, during therapy, diversity may remain high at the genetic level but converge to a single, potentially targetable phenotype, as recently observed in solid cancers [172]. High-throughput and combinatorial functional screens will also be necessary to determine the relative contribution of different evolutionary processes (selection, neutral drift, cooperation) to leukemic transformation and treatment resistance. These steps will be crucial for the development of evolutionary informed therapies. Notably, task repartition might be amenable to evolutionary steering toward evolutionary traps by exploiting fitness trade-offs. The development of such single-cell approaches and the availability of an increasing therapeutic armamentarium will then open a new era for AML, where a deeper understanding of the evolutionary dynamics underlying leukemic transformation and drug resistance will instruct the design of novel biomarkers and treatment strategies, improving patient outcome.

## Figures and Tables

**Figure 1 cancers-13-04887-f001:**
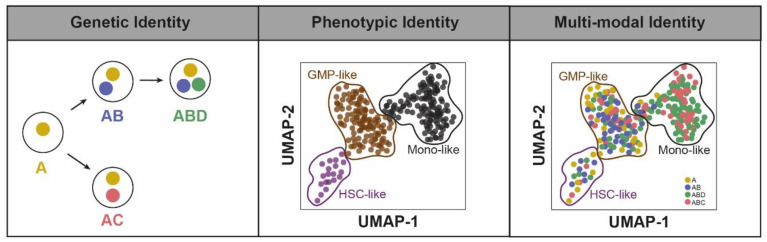
Possible definitions of clonal identity in leukemia.

**Figure 2 cancers-13-04887-f002:**
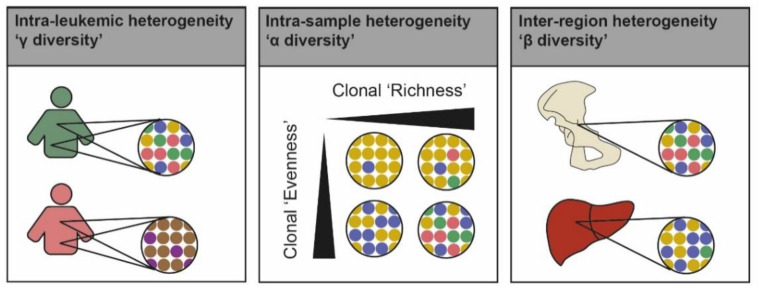
Sources of intraleukemic diversity.

**Figure 3 cancers-13-04887-f003:**
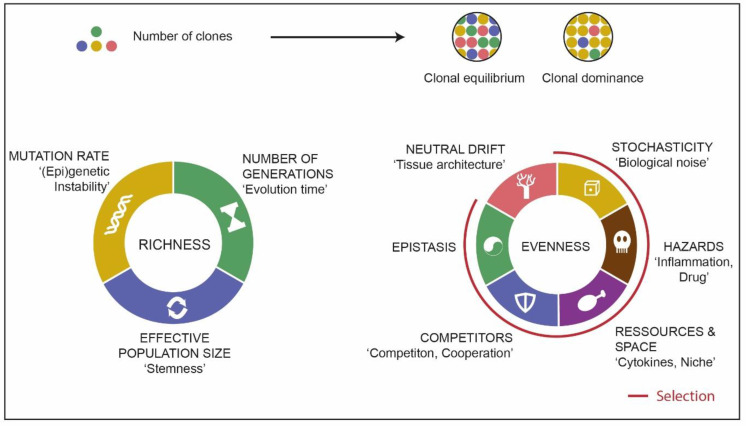
Factors contributing to clonal richness and evenness in leukemia.

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
