# Peer review of "Clonal Architecture and Evolutionary Dynamics in Acute Myeloid Leukemias"

_cancers, 2021, doi:10.3390/cancers13194887_

Round 1

Reviewer 1 Report

This is a rather interesting review article focusing on clonal diversity, namely richness (number of leukemic clones) and evenness (distribution of clone sizes), and further discussing the mechanisms at the origin of clonal emergence and the causes of clonal dynamics in AML.

However, AML is a heterogeneous disease including de novo variant as well as secondary AML (therapy related and MDS-related variants). Secondary AML usually display an higher number of gene anomalies and clonal richness. This information should be briefly mentioned in the summary section, and clonal diversity between the two forms (primary vs secondary variants) clearly documented.

Furthermore, the most important genes and methodology proposed by authors for addressing clonal diversity should be mentioned in the appropriate sections of the manuscript, including the abstract.

The prognostic impact from this analysis is limited since a few research laboratories may have this information available. Readers may be aware of this consideration.

Author Response

Reviewer 1

This is a rather interesting review article focusing on clonal diversity, namely richness (number of leukemic clones) and evenness (distribution of clone sizes), and further discussing the mechanisms at the origin of clonal emergence and the causes of clonal dynamics in AML.

We thank the reviewer for his overall appreciation of the manuscript and his proposals to improve the text (see below).

However, AML is a heterogeneous disease including de novo variant as well as secondary AML (therapy related and MDS-related variants). Secondary AML usually display an higher number of gene anomalies and clonal richness. This information should be briefly mentioned in the summary section, and clonal diversity between the two forms (primary vs secondary variants) clearly documented.

We have added a sentences on this point (lines 195-197 of the revised manuscript).

Furthermore, the most important genes and methodology proposed by authors for addressing clonal diversity should be mentioned in the appropriate sections of the manuscript, including the abstract.

We have added a sentence on this point (lines 91-95 of the revised manuscript).

The prognostic impact from this analysis is limited since a few research laboratories may have this information available. Readers may be aware of this consideration.

We have added a sentence on this point (lines 358-361 of the revised manuscript).

Reviewer 2 Report

The manuscript is a well-written review , although without an original research contribution by the authors. Even so, the review is up-to-date and very comprehensive, in fact on the dense side. I in my opinion it would make the reader more interested if instead of quoting conclusions from studies the authors would actually discuss results of important studies.

Author Response

The manuscript is a well-written review , although without an original research contribution by the authors. Even so, the review is up-to-date and very comprehensive, in fact on the dense side. I in my opinion it would make the reader more interested if instead of quoting conclusions from studies the authors would actually discuss results of important studies.

We thank the reviewer for his overall appreciation of the manuscript. Indeed the manuscript was intended as a review article and does contain additional original research. We believe however that this is compensated by the broader angle chosen to discuss clonal identity beyond genetics.

Reviewer 3 Report

This is a very authoritative manuscript on this topic. 

There are many grammatical and typographical errors that make the paper a little difficult to read (easily correctable)

Line 16: a word is missing after epigenetic (alterations? Or something like that)

Line 17 would read better:  …sequencing, and more recently of single-cell technologies, …

Line 17: “into” should be “onto”

Line 18: should read acute myeloid leukemia (no capitols)

Line 19: should read “…of leukemia cells and discuss…”

Line 22: should read “…neutral drift, and cell-intrinsic…”

Line 23: “patients” should be “patients’ “

Line 24: should read “…design of novel…”

Line 26: a word is missing after epigenetic (alterations? Or something like that)

Line 27: would read better:  …sequencing, and more recently of single-cell technologies, …

Line 28: “into” should be “onto”

Line 29: should read acute myeloid leukemia (no capitols)

Line 30: should read “…of leukemia cells and discuss…”

Line 48: “populations” should be “population”

Line 50: should read acute myeloid leukemia (no capitols)

Line 93: “hands” should be “hand”

Line 188: should “…read heterogeneity, and is…”

Line 147: should read “driver mutations”

Line 157: “stratified” should be “stratify”

Line 182: “mutations” should be “mutation”

Line 183: “alterations” should be “alteration”

Line 185: should read “…with germline mutations…”

Line 189: remove space between references

Line 203: “tissues” should be “tissues’ “

Line 214: “suggest” should be “suggests”

Line 288: “is” should be “are”

Line 296: should read “…and could contribute…”

Line 311: delete excessive spacing

Line 326: “ported should be “portend”

Line 338 should read “…carried out…”

References:

There are formatting errors throughout

Author Response

We thank the reviewer for his overall appreciation of the manuscript. We have corrected all grammatical and typographical errors and apologize for the inconvenience.